# Prevalence of Hepatitis C viral infection in Ghana: A systematic review and meta-analysis protocol

Peter Kwabena Fosu[1]*, Gill ten Hoor[2], Charles Ampong Adjei[3], Fidelis Atibila[4], Robert A. C. Ruiter[5]

1 Public Health Specialist, Department of Medicine, The Trust Hospital, Accra, Ghana, 2 Department of Works and Social Psychology, Maastricht University, Maastricht, the Netherlands, 3 Department of Public Health Nursing, School of Nursing and Midwifery, College of Health Sciences, University of Ghana, Accra, Ghana, 4 University of Hertfordshire, Centre for Postgraduate Medicine and Public Health, College Lane Campus, Hatfield, United Kingdom, 5 Department of Work and Social Psychology, Maastricht University, Maastricht, the Netherlands

* pkfosu@gmail.com

## Abstract

### Background

Hepatitis C virus (HCV) infection remains a major public health concern for many countries. A recent survey report in Ghana revealed a national HCV prevalence rate of 4.6% in a population of 35 million but with notably higher regional variations ranging from 8.6 to 14.4%. Considering that Ghana is targeting micro-elimination of HCV as part of the STOP Hepatitis C project, it is prudent to estimate the current epidemiological burden of hepatitis C for evidence-based policymaking, public health research, and program direction. An initial search of the literature showed a previous review that spanned from 1995 to 2015. The gap of almost 10 years may not reflect the current burden of hepatitis C in Ghana, hence this review. A systematic literature search will be performed in the major electronic databases and search engines including PubMed, Embase, Web of Science, CINAHL, and African Journals Online (AJOL). There will be a search for articles reporting on the prevalence of hepatitis C in Ghana from 2016 to 2024 in these databases. The protocol is registered with PROSPERO (CRD42024592505).

## Introduction

Hepatitis C virus (HCV) infection remains a major public health concern for many countries [1–4]. The HCV affects the liver and mostly occurs through exposure to blood from unsafe injection practices, unsafe medical procedures, HCV infected blood for transfusions, and injection drug use just to mention a few [4]. Unlike hepatitis B virus infection that is preventable, there is currently no effective vaccine against hepatitis C virus. However, direct-acting antiviral medicines (DAAs) can cure more than 95% of persons with hepatitis C [4].

Globally, about 50 million people are living with chronic HCV infection, with over 1.0 million new infections occurring every year [4]. Some of the complications of HCV infections

**Data availability statement:** This is a systematic review protocol. There is currently no data available. Research data will be made publicly available when the study is completed and published.

**Funding:** The author(s) received no specific funding for this work.

**Competing interests:** The authors have declared that no competing interests exist.

include ascites, liver cirrhosis, and liver cancer [4,5]. Out of the 1.3 million hepatitis-related deaths per year, 17% is attributable to chronic HCV infection [4]. There are however regional variations in the prevalence of HCV with countries in the World Health Organization (WHO) Africa region having the highest burden of over 8 million [4]. For example, review reports in Africa document HCV prevalence of 1.65% in Burkina Faso [6], 3.1% in Ethiopia [7], 5.7% in Rwanda [8], 15.2%, 6.6%, and 13.8% among patients, healthcare workers, and the general population in Nigeria respectively [9,10].

In Ghana, a recent nationwide survey revealed a national HCV prevalence rate of about 4.6% in a population of 35 million [11]. There are however regional variations in the prevalence rate of HCV with the five regions in the Northern sector having the highest seroprevalence range between 8.6–14.4% [11]. Some traditional practices, including facial and other body parts scarification performed during the early weeks of life for family and tribal identification, spiritual fortification, and traditional medicine use are contentiously attributed to the high rate of infection in the northern zone as compared to southern Ghana [12].

Given the aforementioned burden of HCV, intervention such as the STOP hepatitis C Ghana project ought to be driven by epidemiological data. Although several studies have previously documented the burden of HCV in Ghana [13–17] most of these have been done at the district or regional levels or even among smaller groups like blood donors [18–22], pregnant women and children [23–26], and people living with HIV and hepatitis B [24,27–31]. Only one study systematically reviewed the burden of HCV in Ghana about a decade ago (1995–2015) [32]. Considering that Ghana is targeting micro-elimination (i.e., focusing intervention on high-risk and vulnerable populations) of HCV, current data on the disease prevalence is crucial for evidence-based policymaking, public health research, and program direction. Findings will contribute to the design of public health response towards hepatitis C elimination in Ghana. This study aims to synthesize evidence on the prevalence of HCV in Ghana from 2016 to 2024.

## Materials and methods

The Joanna Brigg's Institute methodology for systematic reviews [33] will be followed. This review will be designed in line with the internationally Preferred Reporting Items for Systematic Reviews and Meta-analyses (PRISMA) protocols. It will follow the Meta-analysis of Observational Studies in Epidemiology (MOOSE) approach [34–36]. In addition to this, the protocol for this review has been registered in the PROSPERO database (CRD42024592505). There is no need for ethical clearance since only secondary data will be used as pertains in systematic review.

### Review questions

1. What is the prevalence of hepatitis C infection in Ghana?

2. Are there variations in hepatitis C burden among sub-groups, including blood donors, pregnant women, people living with HIV, People with liver diseases, sex workers, injection drug users, and men who have sex with men?

3. What extent does hepatitis C prevalence differ among the sixteen regions of Ghana?

### Inclusion and exclusion criteria

The prevalence of hepatitis C in both the general population and different population strata will be the key sources of data included in this study. Published articles and follow-up studies

with an outcome related to hepatitis C prevalence in the Ghanaian population that were published between 2016–2024 will be eligible for inclusion in this review. Additionally, conference presentations that provide sufficient information about the prevalence of hepatitis C in Ghana will be considered for eligibility. There will be consideration for only articles published in the English language. Multicenter research involving Ghanaian participants across various regions and districts will be included only when there is sufficient statistical information on the prevalence of HCV in Ghana. Prevalence studies done in various minority groups in Ghana such as blood donors, injection drug users (IDU), barbers and nail cutters, healthcare workers, pregnant women and children, and persons living with hepatitis B, HIV, and syphilis and men who have sex with men will also be included.

Exclusion criteria will consist of clinical trials, case-control studies, qualitative or intervention studies, case reports, and case series that do not report absolute population or sub-population prevalence estimates of HCV. Furthermore, editorials, commentary, letters to the editor, author replies, animal studies, or review or modelling studies that do not provide original HCV prevalence outcomes will be excluded.

## Search strategy

A systematic literature search will be performed in the major electronic databases and search engines PubMed, Embase (via Ovid), Web of Science, CINAHL, and African journals online (AJOL). For grey literature, the authors will scrutinize the results in advanced Google Scholar searches in addition to local thesis repositories. Furthermore, a snowballing technique will be applied to analyze the bibliographies of all the eligible studies to identify additional notable publications [37]. A revised approach with search terms specifying and targeting the defined population, intervention (not applicable in this case), defined outcome of interest, and required study setting will be used to screen all study titles and abstracts [35] The search strategy is to use a combination of keywords such as "prevalence", "burden", "hepatitis C", "HCV" and "Ghana" linked by the Boolean operators "OR" and "AND" in our search for the articles ("Prevalence" OR "Burden") AND ("Hepatitis C" OR "HCV") AND "Ghana") [38].

The Database of Abstracts of Reviews of Effects (DARE) (http://www.library.UCSF.edu) and PROSPERO databases will be searched to check for the presence of similar articles related to the topic even before starting the journal search. The literature search, eligible studies selection, data extraction, data analysis, and result reporting will be done per the PRISMA guidelines [35]. The searched articles will be imported to Mendeley Desktop 1.19.4, followed by the identification and removal of all duplicate reports originating from multiple sources [36]. Two independent co-authors will access and pre-screen the articles and abstracts returned from the database queries to flag non-suitable or extraneous reports for exclusion. Subsequently, full-text reports of the pre-screened and selected studies will be retrieved and scrutinized by the two co-authors who will work independently to ensure that there is conformity to the stated inclusion criteria. Any identified inconsistencies in the outcome of the independent screening will be resolved by consensus between the two authors. Relevant data from selected studies will be extracted into a standardized Microsoft Excel template. Per the PRISMA guidelines, the basis for excluding any article during the screening process will be documented and reported [36].

## Data extraction

The following relevant data will be retrieved from selected full-text articles using a standardized extraction form [38,39]. Details of publication such as study title, author names, year of publication, digital object identifiers (DOIs), study setting, study objectives, study design,

study population and sample size, and key findings will be documented. When one study has multiple publications, only the most informative manuscripts will be retained. Moreover, where prevalence estimates were reported for the same population using the same methods across multiple time points, only data for the most recent time point will be extracted [40]. All studies will be assessed for potential risk of bias using the Joanna Briggs Institute validated tool [33]. The purpose of this appraisal is to determine the methodological quality of a study and to assess the extent to which the study has addressed the potential risk of bias in its design, conduct, and analysis.

All articles selected for inclusion in the systematic review will be subjected to rigorous appraisal by two critical independent appraisers who will indicate an overall appraisal result of 'include', 'exclude' or 'seek further information'. The assessment will focus on several key areas, such as the clarity of the criteria for sample inclusion, the detailed description of the study participants and setting, the validity and reliability of the exposure measurement, the use of objective, standard criteria for condition measurement, and the identification and mitigation of confounding factors. Where there is disagreement, consensus will be built among the authors. Reports with scanty details will be classified as "limited," and the primary investigators will be contacted for additional information.

## Meta-analysis / data analysis

Meta-analyses for the prevalence of Hepatitis C Virus (HCV) in Ghana were conducted using random-effects models with a weighted mean difference (WMD) in Review Manager [41].

The $I^2$ test statistics will be used to estimate the presence of observed differences among studies due to heterogeneity. The value can range from 0 to 100%, with 0% indicating the absence of heterogeneity, and 100% corresponding to a significant heterogeneity. The 25%, 50%, and 75% values will represent low, medium, and high heterogeneities between studies, respectively [42]. The presence of heterogeneity will also be determined at a 95% confidence interval (p-value of $< 0.05$) [43]. The possible sources of heterogeneity will be investigated through sensitivity analysis, subgroup analysis, and meta-regression. The presence of publication bias and small study effects will be investigated using visual inspection of funnel plots and Egger's weighted statistics. All the data management and statistical analysis will be performed using random-effects models with a weighted mean difference (WMD) in Review Manager [41].

## Supporting information

**S1 File. Prisma guidelines.**
(DOCX)

**S2 File. The Joanna Brigg's Institute appraisal checklist.**
(DOCX)

## Author contributions

**Conceptualization:** Peter Kwabena Fosu, Gill Ten Hoor, Charles Ampong Adjei.

**Data curation:** Peter Kwabena Fosu, Gill Ten Hoor, Charles Ampong Adjei, Fidelis Atibila, Robert A. C. Ruiter.

**Formal analysis:** Peter Kwabena Fosu, Charles Ampong Adjei, Fidelis Atibila, Robert A. C. Ruiter.

**Methodology:** Peter Kwabena Fosu, Gill Ten Hoor, Charles Ampong Adjei, Fidelis Atibila.

**Supervision:** Gill Ten Hoor, Charles Ampong Adjei, Robert A. C. Ruiter.

**Writing – review & editing:** Peter Kwabena Fosu, Gill Ten Hoor, Charles Ampong Adjei.

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
