## [Decision Letter · Decision Letter 0]

20 Feb 2025

PONE-D-24-49434Prevalence of Hepatitis C Viral Infection in Ghana: A Systematic Review and Meta-Analysis ProtocolPLOS ONE

Dear Dr. Fosu,

Thank you for submitting your manuscript to PLOS ONE. After careful consideration, we feel that it has merit but does not fully meet PLOS ONE’s publication criteria as it currently stands. Therefore, we invite you to submit a revised version of the manuscript that addresses the points raised during the review process.

Your manuscript was reviewed by two experts in the field. Both found some problems in your submission, which require your attention. Please consider the attached comments and provide point-by-point responses.

We look forward to receiving your revised manuscript.

Kind regards,

Yury E Khudyakov, PhD

Academic Editor

PLOS ONE

Journal Requirements:

2. Thank you for stating the following in your Competing Interests section: Authors have no competing interest 

Reviewers' comments:

Reviewer's Responses to Questions

**Comments to the Author**

1. Does the manuscript provide a valid rationale for the proposed study, with clearly identified and justified research questions?

Reviewer #1: Yes

Reviewer #2: Yes

2. Is the protocol technically sound and planned in a manner that will lead to a meaningful outcome and allow testing the stated hypotheses?

Reviewer #1: Yes

Reviewer #2: Yes

3. Is the methodology feasible and described in sufficient detail to allow the work to be replicable?

Reviewer #1: Yes

Reviewer #2: Yes

4. Have the authors described where all data underlying the findings will be made available when the study is complete?

Reviewer #1: No

Reviewer #2: Yes

5. Is the manuscript presented in an intelligible fashion and written in standard English?

Reviewer #1: Yes

Reviewer #2: Yes

6. Review Comments to the Author

You may also provide optional suggestions and comments to authors that they might find helpful in planning their study.

Reviewer #1: The systematic review and meta-analysis protocol for HCV prevalence in Ghana is well written and highlights the need for another prevalence analysis regarding HCV infection in the country.

While there was a multi-centre corss-sectional study for HCV seroprevalence, testing, and treatment capacity in public health facilities in Ghana, 2016-2021 which represents the recent HCV prevalence across different populations and regions, the research question for this protocol is very gereral. I would like to know if the research question would be more specific, such as 2016 to 2024, general population vs sub/key-population, or country vs regional prevalence analysis.

Other than that, I wish all the authers the best for this systematic review and meta-analysis process.

Reviewer #2: The protocol is written fairly well, in line with the standard PRISMA guidelines. Some comments for consideration:

1. Briefly describe the expected outcome of the systematic review and meta-analysis in the abstract.

2. It might help to include a sample search string in an appendix or supplementary file.

3. I would prefer to see a brief discussion section in the protocol that addresses the importance of the topic,

potential limitations, and the expected outcome of the review.

7. PLOS authors have the option to publish the peer review history of their article (what does this mean? ). If published, this will include your full peer review and any attached files.

**Do you want your identity to be public for this peer review?** For information about this choice, including consent withdrawal, please see our Privacy Policy .

Reviewer #1: No

Reviewer #2: No

---

## [Author Response · Author response to Decision Letter 1]

25 Feb 2025

Responds to Editorial comments:

1. We appreciate the editor’s comment. We have ensured that the manuscript meets PLOS ONE’s style requirements. This adjustment can be seen throughout the manuscript.

2. Given that this is a study protocol, we have no data to share at the moment. However, we will make available all the data following the systematic review. This is well captured on page 4.

3. Thank you. All supporting files have been named accordingly, including in text citations. This can be seen throughout the manuscript

4. We have reviewed all the references and they are well structured according to the PLOS ONE’s referencing style. This can be seen on pages 5 to 8.

Responds to Reviewer 1:

1. We appreciate the first reviewer’s positive comment.

2. We appreciate the reviewer’s comment. The manuscript already had the search year range captured on page 6. It reads “There will be a search for articles reporting on the prevalence of hepatitis C in Ghana from 2016 to 2024”. However, we have expanded the research question to include the determination of hepatitis C burden among various sub-groups, including blood donors, pregnant women, sex workers, injection drug users, men who have sex with men, people living with HIV, and people with liver diseases. In addition, the extent of variations of hepatitis C prevalence across the 16 regions of Ghana will be examined. The adjustment can be seen on page 5.

Responds to Reviewer 2:

1. Thank you. We have made an adjustment based on the reviewer’s comment in the manuscript. It now reads “Findings will contribute to the design of public health response towards hepatitis C elimination in Ghana”. This adjustment can be seen on page 3.

2. We appreciate the reviewer’s comment. Although the search strings were captured on page 7 of the manuscript, we have been more explicit and this adjustment could be seen on page 7.

3. We appreciate the reviewer’s comment. Importantly, the relevance of the review is well discussed in the introductory section of the manuscript. Excerpt reads “Considering that Ghana is targeting micro-elimination (i.e. focusing intervention on high-risk and vulnerable populations) of HCV, current data on the disease prevalence is crucial for evidence-based policymaking, public health research, and program direction. We have also added the expected outcome of the review which also reads “Findings will contribute to the design of public health response towards hepatitis C elimination in Ghana”.

We are convinced that this brief discussion is enough for the protocol considering that the actual review report will present more broadly the requested details. This adjustment can be seen on page 3 and 4

---

## [Editor Report · Decision Letter 1]

7 Mar 2025

Prevalence of Hepatitis C Viral Infection in Ghana: A Systematic Review and Meta-Analysis Protocol

PONE-D-24-49434R1

Dear Dr. Fosu,

We’re pleased to inform you that your manuscript has been judged scientifically suitable for publication and will be formally accepted for publication once it meets all outstanding technical requirements.

Kind regards,

Yury E Khudyakov, PhD

Academic Editor

PLOS ONE
---

## [Editor Report · Acceptance letter]

PONE-D-24-49434R1

PLOS ONE

Dear Dr. Fosu,

I'm pleased to inform you that your manuscript has been deemed suitable for publication in PLOS ONE. Congratulations! Your manuscript is now being handed over to our production team.

Kind regards,

on behalf of

Dr. Yury E Khudyakov

Academic Editor

PLOS ONE